# Rank-efficient Mixture of Experts for LLM Finetuning

## Abstract

Large language models (LLMs) have achieved impressive results in many general-purpose domains, but their performance on specific tasks can still be improved through finetuning. Parameter-efficient finetuning (PEFT) aims to tailor an LLM to one or more tasks through a small amount of trainable parameters, requiring reduced computational resources. On the one hand, techniques like low-rank adaptation (LoRA) provide the required parameter efficiency with adapters of low, and fixed, rank, which also limits their flexibility. On the other hand, Mixture of experts (MoEs) enhance the flexibility of a model at the cost of an increased parameter count and computational budget. The combination of the two approaches, parameter-efficient MoEfication, has shown promise in addressing the issues of both. In this work, we propose two methods that improve the rank-efficiency of PEFT adapters, increasing the flexibility and reducing the number of parameters involved in MoEfication. First, SharedLoRA retains the additive nature of LoRA by using a two-tier structure of adapters, thereby **increasing the effective rank of the adapter while also reducing its size**. Second, OperA **replaces additive with quantum-inspired multiplicative interactions** to further drive rank efficiency upwards and number of parameters downwards. We show that both techniques match or surpass the state-of-the-art (SoTA) in its commonly used setup on 6 open-source frontier LLMs and 7 tasks, while using notably fewer parameters. Moreover, we also find that **OperA is optimal given the same parameter budget** for 5 out of 6 models considered, always using fewer parameters than the baseline. Finally, we provide evidence for the superior performance of our methods by analyzing the effective rank of the adapters. Here, our SharedLoRA nearly doubles the rank of the SoTA solution, while our OperA's rank is more than two orders of magnitude greater.

## 1 Introduction

Large language models (LLMs) have demonstrated impressive capabilities (OpenAI, 2025; 2024; Qwen AI, 2025a; Meta AI, 2024) when generalizing to novel tasks, making them the models of choice for a variety of domains (Wang et al., 2024; Liu et al., 2024a; 2023). This success is in part due to the vast datasets with which LLMs are pretrained, granting them general knowledge that is then applied to each specific task. Nonetheless, it has been shown that LLMs can be further specialized (or finetuned) to improve performance in under-represented, and often especially challenging, scenarios (Hu et al., 2022; Parthasarathy et al., 2024). This finetuning process is performed closer to the end application, where data and computational resources are more scarce than in the pretraining phase.

Nonetheless, full finetuning of an LLM still requires massive resources (Xia et al., 2024b) and is unfeasible in many use-cases. Parameter-efficient finetuning (PEFT) has emerged as an effective solution that dramatically reduces the computational costs associated with finetuning LLMs (Han et al., 2024; Hu et al., 2022; Liu et al., 2024b). One of the most successful PEFT paradigms, low-rank adaptation (LoRA; Hu et al. 2022), leverages low-rank approximations to adapt specific subspaces of the full weight matrices, often recovering or even surpassing the performance of its full-rank counterparts. The effectiveness of LoRA is likely a testament to the low-ranked nature of task-specific subspaces in neural networks at large. This hypothesis is corroborated by the relatively poorer performance of LoRA when applied to multiple tasks (Feng et al., 2024; Xia et al., 2024a),

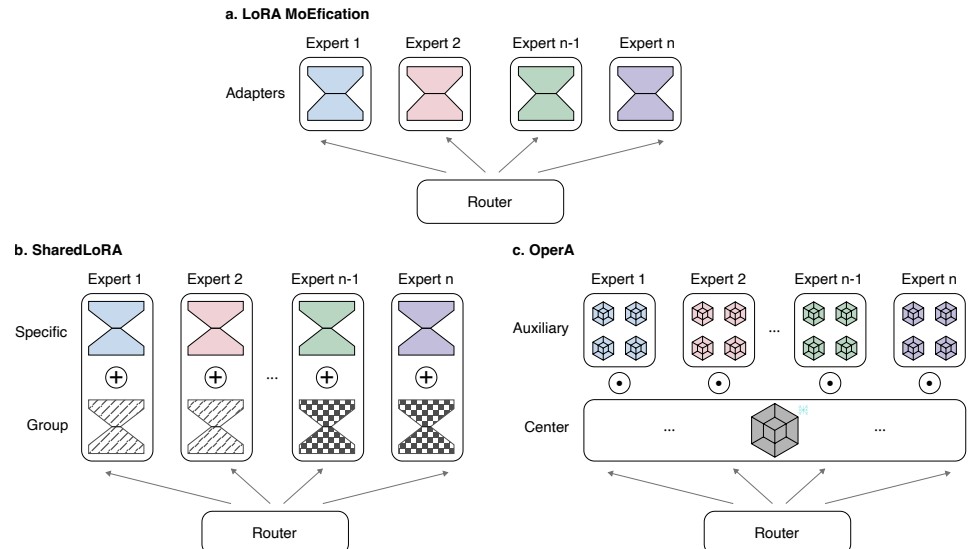

Figure 1: **Rank-efficient Mixture of experts. a.** Regular LoRA MoEfication uses one low-rank adapter per expert. **b.** The experts of SharedLoRA are built by additively combining shared basis LoRAs with expert-specific LoRAs of the same rank. The SharedLoRA rank is therefore the sum of the two ranks. **b.** The experts of OperA are instead built through multiplicative combination of 4D cores, obtained by Matrix product operator (MPO) decomposition.

which can no longer be comfortably represented in low-rank subspaces, but instead require larger solutions.

At the same time, Mixture of experts (MoEs; Shazeer et al. 2017; Fedus et al. 2022) has been widely adopted in recent LLM architectures as a technique to improve performance across tasks without increasing the active parameter count (Jiang et al., 2024; IBM, 2024; DeepSeek-AI, 2025). MoEs leverage multiple linear models (aptly called experts) that process tokens in parallel, offering an attractive factorization of extremely large spaces into more manageable chunks. By instantiating a large number of experts, but only activating few of them for each token, LLMs can be internally specialized without significantly increasing computational costs. However, MoEs still suffer from increased memory requirements, as all experts need to be kept in memory even when not in use.

Finally, LoRAs and MoEs can be combined to obtain a PEFT approach that is parameter-efficient, memory-efficient, and is highly flexible (Tian et al., 2024; Li et al., 2024; Zeng et al., 2025). In particular, each expert is represented as a low-rank linear model, while maintaining the same gating and routing structure of regular MoEs (see Figure 1a), which have already been shown to be sufficiently powerful. Taken individually, each expert can only operate on a restricted subspace, and is therefore susceptible to the same weakness as LoRAs. However, the combination of multiple experts raises the flexibility of the overall model, thereby increasing performance.

We propose here two techniques that **both increase the *effective rank* (Roy & Vetterli, 2007) of the parameter-efficient MoE layer without increasing its parameter count**. Indeed, while the matrix rank might not be an adequate measure of the actual size of the subspace spanned by an adapter due to both practical and numerical reasons, the effective rank provides a more robust measure of the dimensionality of a matrix. First, we introduce SharedLoRA (see Figure 1b). SharedLoRA acts on the rank of the mixture itself, factorizing each expert into two tiers of adapters: (a) specific and (b) grouped. The specific adapters are entirely analogous to conventional low-rank adapters, with one adapter per desired expert. The grouped adapters instead form shareable bases that get added to the specific adapters to increase variety. The combination of specific and grouped adapters results in a reduction in the LoRA rank of each adapter, but in an increase in the rank of each expert.

The second technique we introduce does away with fixed-(and low-)rank adapters altogether by leveraging multiplicative interactions instead of additive ones. To do so, we borrow Matrix product operators (MPOs; Pirvu et al. 2010; Schollwöck 2011; Orús 2014) from the quantum physics liter-

ature to obtain OperA (see Figure 1c). MPOs provide an efficient multiplicative decomposition of arbitrary matrices (originally, Hamiltonians) into *core* and *auxiliary* 4D tensors. By choosing the MPO configuration carefully, we can concentrate most of the weights onto the core tensor and leave it fixed, while learning the leaner auxiliary tensors during training.

We evaluate SharedLoRA and OperA on a comprehensive set of 7 tasks, including ARC (Clark et al., 2018), PIQA (Bisk et al., 2020), OpenBookQA (Mihaylov et al., 2018), BoolQ (Clark et al., 2019), HellaSwag (Zellers et al., 2019), and WinoGrande (Sakaguchi et al., 2021), comparing them with MixLoRA (Li et al., 2024), the current openly available state-of-the-art (SoTA) technique for LoRA MoEfication. We test all approaches on multiple models of different sizes and from different sources to ensure the consistency of our evaluation, including Llama3, Llama3.1, Llama3.2 (Meta AI, 2024), Phi4 (Abdin et al., 2024), and Qwen2.5 (Qwen AI, 2025b). We find that our techniques achieve equal or superior results to SoTA techniques like MixLoRA (Li et al., 2024) and notably increase the effective rank as well, all while reducing the required number of parameters by 38% (SharedLoRA) and 62% (OperA) on average. Moreover, OperA achieves higher accuracy on 5 out of 6 models when given the same parameter budget.

## 2 RELATED WORKS

**Parameter-efficient finetuning (PEFT).** Modern open-weights frontier LLMs can reach tens of billions to hundreds of billions of parameters. Fully finetuning such models for downstream tasks can prove unfeasible in most situations, where parameter-efficient finetuning (PEFT) becomes a necessity. The most popular approach to PEFT for LLMs is LoRA (Hu et al., 2022), which leverages low-rank matrices to finetune a low-dimensional subspace of the original weights to improve performance while reducing the required number of parameters. Further gains on parameter efficiency have been obtained by weight sharing (VeRA; Kopiczko et al. 2024), weight tying (Tied-LoRA; Renduchintala et al. 2024), quantization (QLoRA; Dettmers et al. 2023), and other techniques (Yang et al., 2025; Hayou et al., 2025). The downstream performance of LoRA has also been improved upon, especially through normalization (DoRA; Liu et al. 2024b).

**Mixture of experts (MoEs).** MoEs (Shazeer et al., 2017; Fedus et al., 2022) allow LLMs to efficiently scale the number of parameters without proportionally scaling their compute budgets. In particular, MoEs are part of the conditional computing paradigm, where subsets of the full model are activated in an input-dependent manner. There are multiple variants of MoEs, especially depending on the routing (Fedus et al., 2022; Ruiz et al., 2021; Yue et al., 2025) and the gating mechanism (Li et al., 2023; Puigcerver et al., 2024). Recent works (DeepSeek-AI, 2025; Qwen AI, 2025a) have also shown that very large mixtures with few experts active at a time show remarkable performance across a wide variety of tasks.

**Parameter-efficient MoEfication.** The combination of PEFT with MoEs has recently started gaining traction (Dou et al., 2024; Tian et al., 2024; Li et al., 2024; Zeng et al., 2025), to address the limitations of both approaches. Most parameter-efficient MoEfication techniques (Luo et al., 2024; Gao et al., 2024) replace all linear layers with MoEs. Even greater efficiency is achieved by splitting the LoRA adapters into heads (Tian et al., 2024), driving further factorization. MixLoRA (Li et al., 2024) selectively replaces the QKV matrices in the attention layers with simple LoRAs, while fusing the MLP layers with parameter-efficient MoEs. More recently, S'MoRE (Zeng et al., 2025) proposes hierarchical mixtures with structured graphs. We adopt the architecture of MixLoRA, as it has been shown to be highly effective, and aim to improve the rank efficiency of each adapter. Moreover, in contrast to S'MoRE, our methods (specifically, SharedLoRA) do not introduce graph overheads, but make use of a simpler two-tier structure while obtaining SoTA performance.

**Multiplicative adapters.** Multiplicative adapters for PEFT have seen some adoption, for example with Kronecker-Product (Yu et al., 2025). We explore here Matrix product operators (MPOs), a particularly effective multiplicative decomposition from quantum physics (Pirvu et al., 2010; Schollwöck, 2011; Orús, 2014). MPOs have been used for compressing linear layers (Gao et al., 2020) and as a direct alternative to LoRAs (Liu et al., 2021). More similar to this line of work, MPOE (Gao et al., 2022) also propose an MPO-based approach to parameter-efficient MoEfication. MPOE decompose the weights using MPO, and then finetune both the auxiliary and the core (only

in some steps) tensors. Each expert is then recovered by contracting the shared core tensor with the specific auxiliary tensors. The key differences between MPOE and our OperA lie in both the construction of the experts and the routing mechanism. In particular, OperA does not train the core tensor, making it much more parameter-efficient. Moreover, OperA keeps the residual path as the shared expert, and fuses the adapters with the weights.

# 3 BACKGROUND

**Low-rank adaptation (LoRA).** LoRA is based on low-rank approximations of a matrix. In particular, given a pretrained weight $W \in \mathbb{R}^{M \times N}$, LoRA consists of two matrices $A \in \mathbb{R}^{M \times r}$ and $B \in \mathbb{R}^{r \times N}$, with $r \ll M, N$ being the LoRA *rank*. Crucially, while the LoRA rank places an upper bound on the dimensionality of the adapter, the actual subspace spanned by the adapter might be notably smaller. We explore this effect in detail through the use of the *effective rank* in Section 5.4.

Next, the product $AB \in \mathbb{R}^{M \times N}$ results in a shape-compatible matrix with $W$ that only requires a fraction of its parameters to store, producing considerable space savings. This comes to cost of the rank of $AB$, which is limited to $r$, and therefore to its flexibility. The final weight matrix becomes then

$$W' = W + AB \tag{1}$$

**Mixture of experts (MoEs).** MoEs are an architectural design that increases the flexibility and expressivity of LLMs, while maintaining a constrained increase in the number of parameters. Experts are matrices $E_1, ..., E_n \in \mathbb{R}^{M \times N}$ that independently operate on a given input vector $\boldsymbol{x} \in \mathbb{R}^M$, factorizing a large representation space into multiple additive components. The most popular implementation of MoEs uses a noisy top-k routing mechanism that selects $k$ out of $n$ experts for each $\boldsymbol{x}$, resulting in considerable computational savings at the cost of memory. Specifically, given a router $R$

$$y = \sum_i^n R(\boldsymbol{x})_i E_i(\boldsymbol{x}) \tag{2}$$

with

$$R(\boldsymbol{x})_i = \begin{cases} r_i & \text{if expert } i \text{ has been selected} \\ 0 & \text{otherwise} \end{cases} \tag{3}$$

and $\sum_i^n r_i = 1$ are the routing weights, usually enforced via a *softmax* operation.

**Matrix product operators (MPO).** MPOs, in the setup used in this work, form a multiplicative representation of an arbitrary matrix $W \in \mathbb{R}^{M \times N}$ into 4D tensors (or cores) with a specific configuration. Specifically,

$$W = L_1, L_2, ..., L_n C R_1, R_2, ..., R_n \tag{4}$$

with $L_i, \ i \leq n$ being the left auxiliary cores, $C$ being the center core, and $R_i, \ i \leq n$ being the right auxiliary cores. The left and right auxiliary cores are analogous; we will refer to them simply as auxiliary cores from now on. Therefore, specifying a decomposition with $n$ left (or right) auxiliary cores results in a total of $2n + 1$ tensors. Each tensor has shape $\mathbb{R}^{d_{in}, r, c, d_{out}}$, with $r, c$ being the *physical row resp. column legs* and $d_{in}, d_{out}$ being the *bond dimensions*. The specification of $r$ and $c$ for each core represents the main hyperparameter choice of the MPO system, as we do not restrict the bond dimensions here (while it is more typical in quantum physics applications; Cui et al. 2015). Finally, a straightforward tensor contraction of all cores recovers the original matrix.

# 4 RANK-EFFICIENT FINETUNING TECHNIQUES

We now present the main contributions of this work, SharedLoRA and OperA. We build both with the aim of increasing the effective rank, first by improving additive adapters with SharedLoRA, and then enhancing these adapters with multiplicative interactions with OperA. The two techniques mainly tackle the building of the experts or adapters to obtain greater rank efficiency at lower parameters, while we reuse the SoTA combination of LoRA MoEs with LLMs for PEFT found in MixLoRA (see Appendix B.4 for an ablation of MixLoRA's fusion). In particular, we replace the MLP layers with our MoE adapters and use standard LoRAs for the attention layers.

### 4.1 SHAREDLoRA: TWO-TIER ADDITIVE ADAPTERS

SharedLoRA uses a combination of two tiers of low-rank adapters to obtain experts with a higher effective rank without impacting the parameter count. Given a weight matrix $W$ and $n$ desired experts, we first initialize $n$ first-tier (or specific) adapters $A_i^s, B_i^s,\ i \leq n$ of rank $r$. Next, we initialize $g$ second-tier (or group) adapters $A_i^b, B_i^b,\ i \leq g$, with $g$ a divisor of $n$ such that $n = g \times m$. We then assign the first group adapters $A_1^b, B_1^b$ to the first $m$ experts, the second group adapters to the second $m$ experts, and so on. Next, each expert $E_i$ is defined as

$$E_i = A_i^s B_i^s + A_{i/m}^b B_{i/m}^b \tag{5}$$

Together with its noisy top-k router, the input ($\boldsymbol{x}$)-dependent SharedLoRA layer becomes

$$W' = W + \sum_i^n R(\boldsymbol{x})_i (A_i^s B_i^s + A_{i/m}^b B_{i/m}^b) \tag{6}$$

as is usual in MoEs.

Each $E_i$ has a maximum rank of $2r$, for a total number of parameters of $(n + g) \times (M + N) \times r$. In contrast, flat (non-tiered) LoRA MoEs require $2 \times n \times (M + N) \times r$ parameters to achieve the same maximum rank. The two-tier structure results in a $\frac{2n}{n+g} > 1$ (for $g < n$) reduction of parameters at the same rank.

### 4.2 OPERA: QUANTUM-INSPIRED MULTIPLICATIVE ADAPTERS

OperA leverages multiplicative interactions, which are fundamental to the functioning of deep neural networks (Jayakumar et al., 2020). These interactions are described by MPO, and help OperA achieve an even greater effective rank with lower parameter count compared to SoTA additive adapters. In contrast to LoRA, OperA does not use randomly-initialized fixed-rank adapters, but builds arbitrarily ranked adapters directly from the original weight matrix $W$. The first step of OperA is to obtain the MPO representation of $W$ using tensor-train singular value decomposition (TT-SVD; Oseledets 2011). Without loss of generality, consider the case of a decomposition into three cores

$$W = LCR \tag{7}$$

Given $n$ desired experts, we define the multiplicative adapters $L_i, R_i,\ i \leq n$ by initializing them to equal $L, R$. Next, each expert $E_i$ becomes

$$E_i = L_i C R_i \tag{8}$$

with $C$ fixed. Finally, given the same router $R$ as before the input ($\boldsymbol{x}$)-dependent OperA layer becomes

$$W' = \frac{1}{2}W + \sum_i^n R(\boldsymbol{x})_i \frac{1}{2} E_i \tag{9}$$

with the constant $1/2$ necessary to achieve equality at the beginning of training, analogous to setting the $B$ matrix to 0 in LoRA.

The key to the parameter-efficiency of OperA is to concentrate most of the parameters into $C$, which is not trainable. We remark that the main hyperparameters of TT-SVD consist in the sizes of the *physical legs* $r, c$ of the MPO system. In particular, we set row legs $r_1, r_2, r_3$ (with $r_1 \times r_2 \times r_3 = M$) and column legs $c_1, c_2, c_3$ (with $c_1 \times c_2 \times c_3 = N$). With these parameters set, we can now fully describe the shapes of the MPO:

- $L \in \mathbb{R}^{1 \times r_1 \times c_1 \times d_1}$
- $C \in \mathbb{R}^{d_1 \times r_2 \times c_2 \times d_2}$
- $R \in \mathbb{R}^{d_2 \times r_3 \times c_3 \times 1}$

The bond dimensions $d_1, d_2$ are an effect of TT-SVD and cannot be directly controlled, except by cutting the least-relevant singular values, which would result in a lossy reconstruction of $W$ as in regular SVD. Therefore, we can simply choose $r_2 \gg r_{i \neq 2}$ and $c_2 \gg c_{i \neq 2}$ to obtain the desired effect.

The multiplicative adapters $L_i, R_i$ do not direct limit the rank of the expert, allowing the system to exhibit greater variety and flexibility. Nonetheless, the greater parameter efficiency comes at a computational cost, as each expert is now full-rank and must be obtained from a tensor contraction at every step. Therefore, OperA can be seen as a compute-bound but memory-efficient alternative to LoRA adapters, which is highly beneficial for common GPU implementations where compute is abundant and memory bandwidth is often the principal bottleneck.

## 5 RESULTS

Here we report the results of our evaluation, including all relevant ablations for the different parameters of both SharedLoRA and OperA.

### 5.1 EXPERIMENTAL SETUP

**Models.** We compare MixLoRA, SharedLoRA, and OperA on a variety of models to ensure the consistency of our performance results. From Meta, we test: Llama3-8B, Llama3.1-8B, and Llama3.2-3B. From Qwen, we test: Qwen2.5-3B, and Qwen2.5-7B. Finally, from Microsoft we test Phi4-14B. This choice of 6 models covers a representative range of sizes (from 3B to 14B parameters) and of model origins.

**Tasks.** We evaluate our work on multiple datasets and tasks with varying size and complexity. For each dataset, we first finetune each adapter on the training split, and then evaluate the accuracy on the testing split. The task ensemble consists of question-and-answer tasks (ARC-e, ARC-c, PIQA, OpenBookQA), classification tasks (BoolQ), and completion tasks (HellaSwag, WinoGrande). All tasks are evaluated on accuracy.

**MoEs.** For all experiments, we use 8 total experts with 2 active experts per token. Following best practices, we use noisy top-k routing with an auxiliary router balancing loss.

**LoRA.** For both LoRA based approaches, MixLoRA and SharedLoRA, the rank of the adapters is the main hyperparameter. For MixLoRA, the rank is set to either 16 (original, see Section 5.2) or 12 (optimal, see Appendix B.1). For SharedLoRA, we match and surpass the parameter efficiency of MixLoRA with ranks of either 12 or 8.

**MPO.** MPOs, as used in this work, do not have rank. We instead focus on the sizes of the physical legs to achieve the desired parameter efficiency. The sizes are specific to each model, and can be found in Appendix A. In OperA, we also use simple LoRA for the QKV matrices in the attention components, and fix the rank to 6 to give enough flexibility for adaptation without contributing a large amount of parameters. Therefore, most of the finetuning is left to the multiplicative adapters.

**Environment.** All experiments are performed on Nvidia A100 80GB using PyTorch 2.6 and the MoE-PEFT (Li et al., 2024) codebase. All models are trained using mixed fp32 and bf16.

### 5.2 BASELINE

We replicate the setup of MixLoRA to obtain comparable results with the current SoTA, and follow the original hyperparameter selection which found rank 16 to be optimal. In Table 1 we show that SharedLoRA equals or surpasses the baseline with a smaller rank (12 vs 16), leading to 7.1% average parameter savings. At the same time, OperA at SoTA accuracy achieves on average a 36% space saving across all tested models. Overall, SharedLoRA and OperA surpass the baseline on all the models tested, with notable parameter efficiency gains (see Appendix A for the configuration of each adapter).

### 5.3 FIXED PARAMETER BUDGET

We have demonstrated that SharedLoRA and OperA are competitive with the SoTA while requiring notably less parameters. At the same time, in Appendix B.1 we also further optimize MixLoRA

Table 1: **Accuracy of MixLoRA, and our SharedLoRA and OperA**. For each model, we report the reduction in parameters over MixLoRA and the task accuracy. The baseline parameter count is taken as MixLoRA with rank 16.

| Model | Adapter | % redux ↓ | ARC-c ↑ | ARC-e ↑ | PIQA ↑ | OBQA ↑ | BoolQ ↑ | HS ↑ | WG ↑ | Avg ↑ |
|---|---|---|---|---|---|---|---|---|---|---|
| | MixLoRA | -0.0 | 75.76 | 87.08 | 86.18 | 89.00 | 73.61 | 94.63 | 83.11 | 84.20 |
| Llama3-8B | SharedLoRA | -0.0 | 75.51 | 86.28 | 86.02 | 86.60 | 74.13 | 94.03 | 82.79 | 83.62 |
| | OperA | -0.0 | 77.39 | 86.28 | 88.30 | 87.60 | 76.36 | 96.15 | 81.45 | **84.79** |
| | MixLoRA | -0.0 | 76.96 | 86.82 | 86.40 | 83.20 | 74.65 | 94.77 | 84.14 | 83.85 |
| Llama3.1-8B | SharedLoRA | -7.1 | 76.11 | 87.21 | 86.18 | 83.00 | 74.43 | 94.52 | 83.74 | 83.60 |
| | OperA | -28.2 | 76.88 | 86.32 | 87.92 | 85.20 | 75.23 | 96.10 | 80.58 | **84.03** |
| | MixLoRA | -0.0 | 68.26 | 83.12 | 83.02 | 77.40 | 71.50 | 92.13 | 75.14 | 78.65 |
| Llama3.2-3B | SharedLoRA | -7.6 | 69.28 | 83.29 | 84.00 | 79.00 | 72.20 | 91.70 | 76.72 | **79.46** |
| | OperA | -29.8 | 66.98 | 81.73 | 83.19 | 77.20 | 70.98 | 92.93 | 74.19 | 78.17 |
| | MixLoRA | -0.0 | 91.21 | 94.28 | 91.90 | 92.60 | 75.47 | 95.95 | 88.79 | 90.03 |
| Phi4-14B | SharedLoRA | -6.5 | 90.36 | 95.20 | 92.44 | 92.00 | 76.18 | 96.12 | 88.32 | **90.09** |
| | OperA | -45.8 | 88.14 | 92.84 | 91.84 | 91.00 | 74.49 | 95.25 | 86.27 | 87.43 |
| | MixLoRA | -0.0 | 80.12 | 88.64 | 86.62 | 87.60 | 70.92 | 91.67 | 80.03 | 83.66 |
| Qwen2.5-3B | SharedLoRA | -6.9 | 79.95 | 88.05 | 86.23 | 87.40 | 71.47 | 91.96 | 81.21 | **83.75** |
| | OperA | -60.6 | 80.89 | 89.40 | 86.02 | 87.60 | 70.06 | 92.99 | 74.82 | 83.11 |
| | MixLoRA | -0.0 | 87.88 | 94.19 | 88.74 | 92.80 | 74.52 | 95.29 | 83.27 | 88.10 |
| Qwen2.5-7B | SharedLoRA | -7.1 | 87.03 | 92.04 | 88.36 | 92.20 | 74.43 | 94.85 | 85.16 | 87.72 |
| | OperA | -24.1 | 87.71 | 93.39 | 90.42 | 91.40 | 74.74 | 95.91 | 83.66 | **88.18** |

both for accuracy and for parameter count, and show that SharedLoRA and OperA outperform SoTA in this setting as well. However, the question remains whether OperA can actually leverage its multiplicative interactions to be more parameter-efficient than the baseline. Table 2 indicates that OperA is Pareto-optimal on 5 out of the 6 models, achieving higher accuracy at a lower parameter count than the baseline. Phi4 is the only exception, possibly due to its greater size which hampers convergence.

Table 2: **Comparison of MixLoRA and OperA given the same parameter budget.** For each model, we report the reduction in parameters over MixLoRA and the task accuracy. The baseline parameter count is taken as MixLoRA with rank 16.

| Model | Adapter | % redux ↓ | ARC-c ↑ | ARC-e ↑ | PIQA ↑ | OBQA ↑ | BoolQ ↑ | HS ↑ | WG ↑ | Avg ↑ |
|---|---|---|---|---|---|---|---|---|---|---|
| Llama3-8B | MixLoRA | -56.0% | 78.75 | 87.12 | 83.81 | 85.20 | 74.62 | 94.75 | 83.90 | 84.02 |
| | OperA | -60.6% | 77.64 | 86.41 | 87.60 | 84.20 | 74.74 | 95.71 | 83.03 | **84.19** |
| Llama3.1-8B | MixLoRA | -56.0% | 78.33 | 86.45 | 83.60 | 87.40 | 75.69 | 95.16 | 83.42 | 84.29 |
| | OperA | -60.6% | 77.73 | 87.29 | 88.19 | 85.40 | 75.69 | 95.91 | 80.03 | **84.75** |
| Llama3.2-3B | MixLoRA | -55.7% | 69.33 | 83.42 | 83.17 | 79.40 | 70.98 | 92.15 | 76.64 | 79.30 |
| | OperA | -55.0% | 69.08 | 84.64 | 82.64 | 79.40 | 71.53 | 93.60 | 75.16 | **79.44** |
| Phi4-14B | MixLoRA | -71.4% | 91.47 | 96.68 | 92.27 | 93.60 | 76.02 | 95.81 | 87.85 | **90.53** |
| | OperA | -45.8% | 88.14 | 92.84 | 91.84 | 91.00 | 74.49 | 95.38 | 87.21 | 88.54 |
| Qwen2.5-3B | MixLoRA | -55.9% | 82.08 | 88.39 | 85.70 | 84.93 | 70.80 | 92.76 | 79.79 | 83.49 |
| | OperA | -60.6% | 80.89 | 89.40 | 86.02 | 87.60 | 70.06 | 92.99 | 77.82 | **83.54** |
| Qwen2.5-7B | MixLoRA | -56.1% | 86.77 | 91.92 | 88.93 | 92.40 | 75.17 | 95.36 | 85.79 | 88.05 |
| | OperA | -63.2% | 87.63 | 92.51 | 90.59 | 94.00 | 74.07 | 95.76 | 83.98 | **88.36** |

## 5.4 EFFECTIVE RANK

While the LoRA rank puts an upper bound to the dimensionality of the PEFT, whether the adapters do actually make use of all available space can be directly investigated. Here, we choose the effective rank (Roy & Vetterli, 2007) as the measure of interest due to its inherent robustness. The effective

rank is the measure of the entropy of the singular values of a matrix, computed as:

$$\text{erank}(W) = \frac{\|W\|_F^2}{\|W\|_2^2} \tag{10}$$

where $\|\cdot\|_F$ is the Frobenius norm and $\|\cdot\|_2$ is the L2 norm. The effective rank computes the ratio between the sum of all singular values to the greatest one, quantifying the concentration of the singular values and therefore the effective dimensionality of the matrix.

We compute the effective rank of both MixLoRA and SharedLoRA adapters as training progresses as the effective rank of the full LoRA matrix, i.e. $\text{erank}(E_i)$. In the case of OperA, we compute the effective rank of $W - E_i$ instead, as each $E_i$ is always full rank and we wish to evaluate the additional dimensionality brought by $E_i$ to the system, rather than its raw dimensionality.

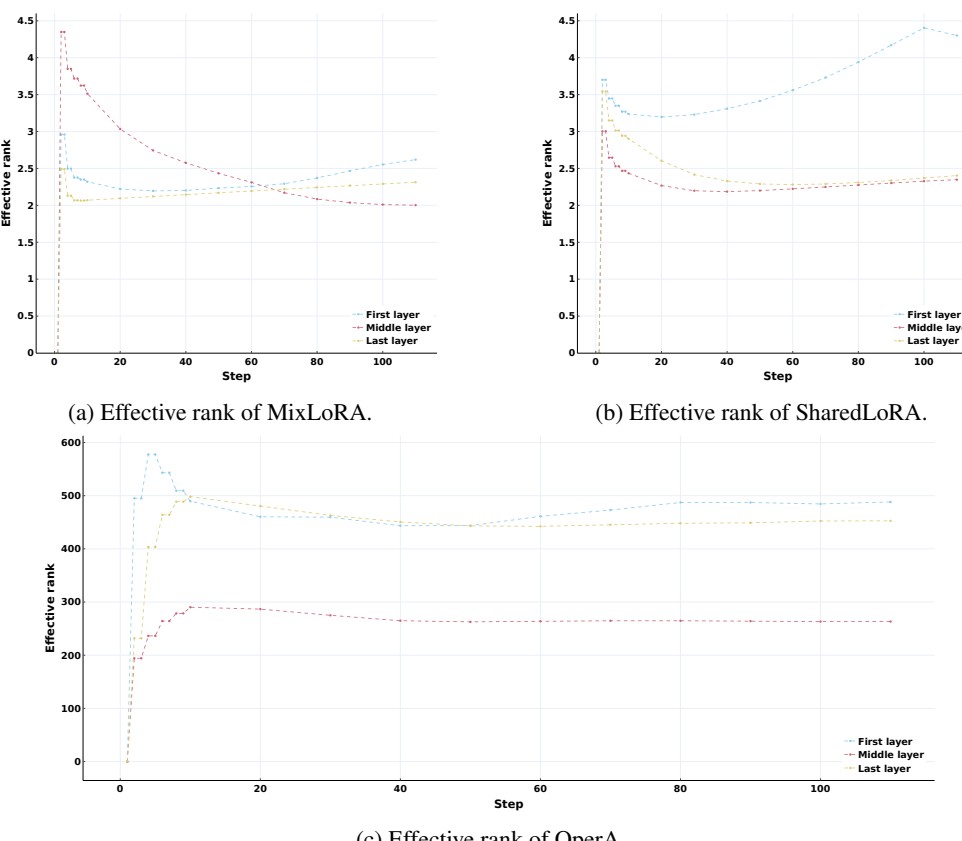

(a) Effective rank of MixLoRA.  (b) Effective rank of SharedLoRA.

(c) Effective rank of OperA.

Figure 2: **Effective rank of the adapters.** Development of the effective rank as finetuning progresses.

Figure 2 shows that the effective rank quickly rises at the beginning of training, as expected, but then progressively settles. In the case of MixLoRA, the effective rank degrades quickly, hovering around 2.5 for both shallow and deep layers, consistent with the findings in Li et al. (2024) that performance is maintained down to a LoRA rank of 2. On the other hand, the effective rank of SharedLoRA remains notably higher, especially in the first layer, surpassing 4. This indicates both that SharedLoRA achieves a higher effective rank, and also that it places more emphasis on shallower layers, closer to the input. Finally, OperA's effective rank is notably higher and more stable than the additive approaches, indicating that the additional information gained by OperA spans a much greater subspace even with fewer parameters.

## 5.5 ABLATIONS

**Optimal rank.** The main hyperparameter that determines the parameter count of LoRA-based approaches is the rank of the adapters. As mentioned earlier, to obtain baseline results we first apply

MixLoRA with rank 16, found to be optimal in the original work. However, we also perform a more fine-grained search using more modern architectures to ensure as strong a baseline as possible. Fig. 3 shows that performance peaks at rank 12 for Llama3-8B, while it remains stable after rank 12 for Qwen2.5-3B. Therefore, we choose rank 12 as the most parameter-efficient solution for both, and extend this evaluation to all architectures to establish an optimized MixLoRA baseline in Appendix B.1.

**Optimal MPO size.** The parameter count of OperA is mainly determined by the configuration of the physical legs, which are specific to each model and are evaluated empirically for effectiveness. Nonetheless, the amount of legs also contributes to the downstream performance by varying the degree of factorization. To obtain the optimal number, we sweep from 3 to 7 legs on Qwen2.5-3B and check the average performance across 5 tasks. Figure 2c shows that 5 legs provides the best accuracy.

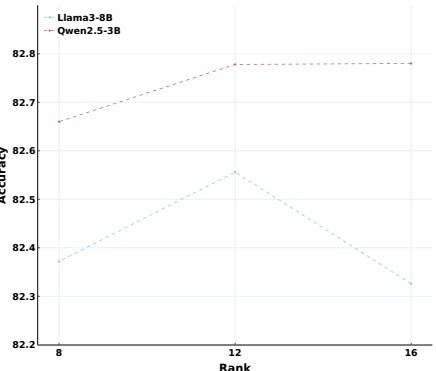

Figure 3: **MixLoRA performance depends on rank.** We report the average accuracy of Llama3-8B and Qwen2.5-3B at varying LoRA ranks, using MixLoRA.

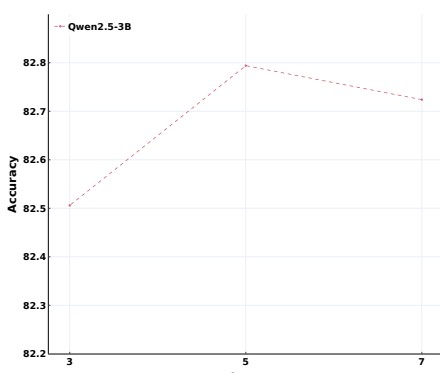

Figure 4: **OperA performance depends on the number of legs.** We report the average accuracy of Qwen2.5-3B at various leg configurations, using OperA.

## 6 CONCLUSION

In this work, we present SharedLoRA and OperA, two rank-efficient techniques for PEFT MoEfication. SharedLoRA is a two-tiered structure of LoRA adapters, which makes use of the more conventional additive interactions. This structure allows SharedLoRA to achieve higher performance at lower parameter counts, by achieving a higher effective rank than the SoTA. OperA, instead, is built on multiplicative interactions, which are fundamental to learning in deep neural networks (Jayakumar et al., 2020). OperA leverages MPOs, a multiplicative decomposition stemming from quantum physics, to produce low-parameter adapters which can be combined with the pretrained weights to obtain powerful high-rank experts. Therefore, we highlight the usefulness of the effective rank as a quantification of the dimensionality of an adapter, and show that targeting it leads to improved performance without an increase in parameter count.

Overall, we show that SharedLoRA and OperA reach SoTA results on most of the models and tasks tested. Specifically, OperA reaches a higher accuracy than the baseline at a lower parameter cost on 5 out of the 6 models tested, making it the optimal choice for a given parameter budget.

Further work is necessary to evaluate the impact of structured compositions of LoRAs, such as our two-tier SharedLoRA and the graph-based S'MoRE Zeng et al. (2025). At the same time, multiplicative adapters Yu et al. (2025) have shown significant promise to replace additive adapters such as LoRA in some situations, with our OperA achieving optimal performance across the board. However, the level of optimization that contributes to the success of LoRA has not yet been achieved for its multiplicative alternatives, and is a significant avenue for future research.

ETHICS STATEMENT

This work does not involve human subjects, personally identifiable information, or sensitive data. All experiments were conducted using publicly available models and datasets. Our focus on improving model performance in low-compute budget environments was a central consideration to mitigate environmental impact. The authors declare no known conflicts of interest.

REPRODUCIBILITY STATEMENT

This paper describes the proposed SharedLoRA and OperA techniques, including the details of the algorithms in Section 4 and the experimental setup in Section 5.1.

The hyperparameters for all the models evaluated can be found in Appendix A.

The code is provided in the supplementary materials, along with the required experimental environments.

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

# A  ADAPTER DETAILS

## A.1  MixLoRA

We report the hyperparameters chosen for MixLoRA in Table 3. The hyperparameters are chosen based on the findings in the original paper, and then further finetuning as described in this work.

Table 3

| Model | Rank | Alpha | Dropout | Results |
|-------|------|-------|---------|---------|
| All | 16 | 32 | 0.05 | Section 5.2 |
| All | 12 | 24 | 0.05 | Appendix B.1 |
| All | 8 | 16 | 0.05 | Section 5.5 |
| All | 7 | 14 | 0.05 | Section 5.4 |

## A.2  SHAREDLORA

We report the hyperparameters chosen for SharedLoRA in Table 3. The hyperparameters are chosen based on an search conducted on the rank and dropout ratio.

Table 4

| Model | Rank | Alpha | Dropout | Results |
|-------|------|-------|---------|---------|
| All | 12 | 24 | 0.2 | Section 5.2 |
| All | 8 | 16 | 0.2 | Appendix B.1 |

## A.3  OPERA

The hyperparameters for OperA depend on the chosen model, specifically the size of the physical legs. We report in Table 5 both the row legs (Row) and the column legs (Column) in the format $r_1 : r_2 : ... : r_l$ and $c_1 : c_2 : ... : c_l$, with $l$ the number of legs. The number of legs here is always 5, as found in Section 5.5.

Table 5

| Model | Column | Row | Rank | Alpha | Dropout | Results |
|-------|--------|-----|------|-------|---------|---------|
| Llama3-8B | 2:8:32:4:2 | 4:7:32:4:4 | 6 | 12 | 0.2 | Section 5.2 |
| | 2:8:32:4:2 | 4:7:32:4:4 | 6 | 12 | 0.2 | Section 5.4, 5.5, Appendix B.1 |
| Llama3.1-8B | 2:8:32:4:2 | 4:7:32:4:4 | 6 | 12 | 0.2 | Section 5.2 |
| | 2:8:32:4:2 | 4:7:32:4:4 | 6 | 12 | 0.2 | Section 5.4, 5.5, Appendix B.1 |
| Llama3.2-3B | 2:8:12:8:2 | 2:8:32:8:2 | 6 | 12 | 0.2 | Section 5.2 |
| | 2:8:12:8:2 | 2:8:32:8:2 | 6 | 12 | 0.2 | Section 5.4, 5.5, Appendix B.1 |
| Phi4-14B | 2:8:40:4:2 | 4:8:35:4:4 | 6 | 12 | 0.2 | Section 5.2 |
| | 2:8:40:4:2 | 4:8:35:4:4 | 6 | 12 | 0.2 | Section 5.4, 5.5, Appendix B.1 |
| Qwen2.5-3B | 2:8:16:4:2 | 2:8:43:8:2 | 6 | 12 | 0.2 | Section 5.2 |
| | 2:8:16:4:2 | 2:8:43:8:2 | 6 | 12 | 0.2 | Section 5.4, 5.5, Appendix B.1 |
| Qwen2.5-7B | 2:8:28:4:2 | 4:8:37:8:2 | 6 | 12 | 0.2 | Section 5.2 |
| | 2:8:28:4:2 | 4:8:37:8:2 | 6 | 12 | 0.2 | Section 5.4, 5.5, Appendix B.1 |

# B    ADDITIONAL RESULTS

## B.1    RANK OPTIMIZATION

We perform an hyperparameter search on Llama3-8B and Qwen2.5-3B to obtain the optimal rank of MixLoRA on more recent architectures. Our rank sweep is more fine grained that in the original work, and our results indicate that substantial space savings can be achieved with rank 12 (see Section 5.4 for the ablation results). Therefore, we also aim to reduce the parameters of the MoEs in SharedLoRA and OperA. The results in Table 6 indicate that both our methods are stable and amenable to different configurations, providing evidence for their flexibility. This is especially remarkable for OperA, which shows similar robustness to an already established method such as LoRA while employing a notably different technique. Specifically, the performance of SharedLoRA and OperA in Table 6 are again equal or superior to MixLoRA, with even more impressive space savings of 38% and 58% respectively over the original baseline.

Table 6: **Accuracy of rank-optimized MixLoRA, SharedLoRA, and OperA**. For each model, we report the reduction in parameters over MixLoRA and the task accuracy. The baseline parameter count is taken as MixLoRA with rank 16.

| Model | Adapter | % redux ↓ | ARC-c ↑ | ARC-e ↑ | PIQA ↑ | OBQA ↑ | BoolQ ↑ | HS ↑ | WG ↑ | Avg ↑ |
|---|---|---|---|---|---|---|---|---|---|---|
| | MixLoRA | -24.9 | 77.13 | 87.84 | 85.90 | 87.20 | 74.71 | 94.70 | 83.35 | **84.40** |
| Llama3-8B | SharedLoRA | -38.2 | 78.92 | 86.24 | 86.02 | 86.60 | 74.13 | 94.92 | 83.19 | 84.29 |
| | OperA | -60.6 | 77.64 | 86.41 | 87.60 | 84.20 | 74.74 | 95.71 | 83.03 | 84.19 |
| | MixLoRA | -24.9 | 76.11 | 85.27 | 87.76 | 84.20 | 75.50 | 94.93 | 84.61 | 84.05 |
| Llama3.1-8B | SharedLoRA | -38.2 | 76.62 | 87.46 | 86.62 | 86.60 | 74.65 | 94.92 | 84.29 | 84.45 |
| | OperA | -60.6 | 77.73 | 87.29 | 88.19 | 85.40 | 75.69 | 95.91 | 83.03 | **84.75** |
| | MixLoRA | -25.2 | 69.28 | 84.85 | 83.57 | 79.80 | 72.20 | 92.53 | 73.09 | 79.33 |
| Llama3.2-3B | SharedLoRA | -38.2 | 68.00 | 82.66 | 83.79 | 79.00 | 72.20 | 92.87 | 76.24 | 79.25 |
| | OperA | -55.0 | 69.08 | 84.64 | 82.64 | 79.40 | 71.53 | 93.60 | 75.16 | **79.44** |
| | MixLoRA | -25.0 | 90.36 | 96.84 | 91.89 | 94.00 | 75.41 | 95.81 | 88.32 | 90.38 |
| Phi4-14B | SharedLoRA | -37.8 | 90.87 | 95.70 | 91.78 | 91.80 | 75.72 | 96.01 | 88.08 | **89.99** |
| | OperA | -71.4 | 88.14 | 92.84 | 91.84 | 91.00 | 74.49 | 95.38 | 87.21 | 88.54 |
| | MixLoRA | -25.0 | 81.06 | 88.59 | 86.56 | 86.40 | 71.28 | 91.94 | 79.08 | 83.56 |
| Qwen2.5-3B | SharedLoRA | -37.8 | 81.57 | 89.31 | 85.96 | 87.60 | 71.56 | 92.16 | 79.32 | **83.93** |
| | OperA | -60.6 | 80.89 | 89.40 | 86.02 | 87.60 | 70.06 | 92.99 | 77.82 | 83.54 |
| | MixLoRA | -24.9 | 87.54 | 92.17 | 88.90 | 92.80 | 74.10 | 95.60 | 84.14 | 87.89 |
| Qwen2.5-7B | SharedLoRA | -37.9 | 87.03 | 91.54 | 89.99 | 92.20 | 73.90 | 95.10 | 84.14 | 87.56 |
| | OperA | -63.2 | 87.63 | 92.51 | 90.59 | 94.00 | 74.07 | 95.76 | 83.98 | **88.36** |

## B.2    RANK ABLATION

Table 7 report all the results of the rank ablation performed on Qwen2.5-3B and Llama3-7B.

Table 7: **Results of the rank ablation**. We sweep the rank of MixLoRA from 16 to 8 on Qwen2.5-3B and Llama3-8B, to determine the optimal configuration.

| Adapter | Model | Rank ↓ | % redux ↓ | ARC-c ↑ | ARC-e ↑ | PIQA ↑ | OBQA ↑ | BoolQ ↑ | Avg ↑ |
|---|---|---|---|---|---|---|---|---|---|
| | | 16 | -0% | 80.12 | 88.64 | 86.62 | 87.60 | 70.92 | **82.78** |
| | Qwen2.5-3B | 12 | -24.5% | 81.06 | 88.59 | 86.56 | 86.40 | 71.28 | **82.78** |
| MixLoRA | | 8 | -49.5% | 81.91 | 89.01 | 83.19 | 87.60 | 71.59 | 82.66 |
| | | 16 | -0% | 75.76 | 87.08 | 86.18 | 89.00 | 73.61 | 82.33 |
| | Llama3-8B | 12 | -24.9% | 77.13 | 87.84 | 85.90 | 87.20 | 74.71 | **82.56** |
| | | 8 | -49.8% | 78.84 | 87.33 | 83.59 | 86.80 | 75.30 | 82.37 |

## B.3 MPO SIZE ABLATION

Table 8 report all the results of the ablation on the number of legs of the MPO performed on Qwen2.5-3B.

Table 8: **Results of the MPO size ablation**. We sweep the number of legs in OperA from 3 to 7 on Qwen2.5-3B, to determine the optimal configuration.

| Adapter | Model | Column | Row | % redux ↓ | ARC-c ↑ | ARC-e ↑ | PIQA ↑ | OBQA ↑ | BoolQ ↑ | Avg ↑ |
|---|---|---|---|---|---|---|---|---|---|---|
| | | 2:2:4:16:2:2 | 2:2:4:43:4:2:2 | -60.6% | 80.55 | 89.52 | 85.53 | 88.20 | 69.82 | 82.72 |
| OperA | Qwen2.5-3B | 2:8:16:4:2 | 2:8:43:8:2 | -60.6% | 80.89 | 89.40 | 86.02 | 87.60 | 70.06 | **82.79** |
| | | 16:16:8 | 16:43:16 | -60.6% | 80.03 | 89.35 | 85.53 | 88.60 | 69.02 | 82.51 |

## B.4 FUSION

The adapter fusion approach of MixLoRA (Li et al., 2024) has been shown to notably improve the performance of additive adapters with LoRA. In Table 9 we report that such approach is helpful for multiplicative adapters such as OperA as well.

Table 9: **Results of the fusion ablation**. We evaluate whether the adapter fusion approach is successful for OperA/ as well on Llama3-8B and Qwen2.5-3B.

| Model | Adapter | ARC-c ↑ | ARC-e ↑ | PIQA ↑ | OBQA ↑ | BoolQ ↑ | Avg ↑ |
|---|---|---|---|---|---|---|---|
| Llama3-8B | OperA (w/o fusion) | 73.46 | 84.30 | 87.11 | 86.6 | 74.07 | 81.11 |
| | OperA | 77.64 | 86.41 | 87.6 | 84.2 | 74.74 | **84.19** |
| Qwen2.5-3B | OperA (w/o fusion) | 76.54 | 87.84 | 82.92 | 85.8 | 69.17 | 80.45 |
| | OperA | 80.89 | 89.40 | 86.02 | 87.60 | 70.06 | **83.54** |

