# OpenReview forum: "Rank-efficient Mixture of Experts for LLM Finetuning"
_ICLR.cc/2026/Conference — ICLR 2026 Conference Withdrawn Submission_

### Official Review · Reviewer_2TKM · 2025-10-31

**Soundness:** 2
**Presentation:** 3
**Contribution:** 3
**Rating:** 2
**Confidence:** 5

**Summary:**

This paper proposes two new rank-efficient PEFT techniques for LLMs: SharedLoRA and OperA. SharedLoRA extends the traditional LoRA framework by introducing a two-tier additive structure consisting of shared group adapters and task-specific adapters, which increases the effective rank of each expert while reducing total parameter count. OperA takes a more radical approach by introducing multiplicative interactions inspired by quantum matrix product operator decompositions, replacing additive adapters with multiplicative tensor cores to achieve even higher rank efficiency. Experiments across six open-source LLMs (Llama3, Qwen2.5, Phi4, etc.) and seven benchmarks (ARC, PIQA, BoolQ, HellaSwag, WinoGrande, etc.) demonstrate that both methods match or outperform SOTA MixLoRA while reducing parameters by 38% (SharedLoRA) and 62% (OperA). The paper also provides an analysis of effective rank, showing that SharedLoRA roughly doubles and OperA increases the rank by two orders of magnitude relative to baseline.

**Strengths:**

1. This paper introduces a two-tier additive adapter structure SharedLoRA that effectively doubles the adapter’s rank while reducing total parameters through shared group bases.
2. OperA leverages quantum-inspired multiplicative adapters using Matrix Product Operator decomposition, allowing high-rank expressivity with minimal trainable parameters.
3. Extensive experiments on six open-source LLMs (Llama3, Qwen2.5, Phi4) and seven benchmarks (ARC-e/c, PIQA, OBQA, BoolQ, HellaSwag, WinoGrande) show that both SharedLoRA and OperA consistently match or surpass MixLoRA. SharedLoRA achieves around 38% parameter savings, while OperA reduces parameters by up to 62% and attains higher accuracy on five out of six models.

**Weaknesses:**

1. The two proposed methods, SharedLoRA and OperA, are conceptually independent and could each warrant a standalone study. Combining them in a single paper slightly dilutes the narrative focus and comparative depth.
2. Finding the optimal MPO configuration for OperA requires extensive hyperparameter exploration, yet the paper provides limited guidance on how to select or approximate the best MPO size across different models.
3. The benchmarks used (ARC, PIQA, BoolQ, HellaSwag, WinoGrande, etc.) are relatively easy to the models; including more recent or challenging benchmarks would better demonstrate generalization and robustness.
4. The paper does not report the baseline performance of the base model of each tested model on these benchmarks, making it difficult to contextualize the improvement magnitude or assess the real transfer efficiency.

**Questions:**

1. How could the two methods be effectively combined into a unified framework, leveraging the additive and multiplicative advantages of both?
2. Under what conditions or model/task characteristics should researchers prefer SharedLoRA over OperA, and vice versa?

---

### Official Review · Reviewer_mAYR · 2025-10-31

**Soundness:** 2
**Presentation:** 2
**Contribution:** 2
**Rating:** 2
**Confidence:** 4

**Summary:**

This paper proposes two new rank-efficient MoE fine-tuning methods.
SharedLoRA employs a two-tier additive structure that enhances the effective rank and reduces parameters through specific and grouped adapters,while OperA, inspired by the Matrix Product Operator (MPO) in quantum physics, decomposes weight matrices via multiplicative factorization.Experimental results show that both methods — especially OperA — outperform MixLoRA in terms of performance.

**Strengths:**

1.	The SharedLoRA method features a simple structure that effectively enhances the representational capacity of LoRA.

2.	OperA significantly reduces the number of trainable parameters while maintaining excellent performance.

3.	The authors explain the superiority of both methods from the perspective of effective rank and provide strong empirical evidence.

**Weaknesses:**

1.	The baseline comparison is quite limited, making it difficult to convincingly demonstrate the effectiveness of the proposed methods. Is MixLoRA truly the state of the art (SoTA)?

2.	While the paper introduces two methods, SharedLoRA and OperA, their conceptual connection and methodological consistency appear weak, which makes the overall framework somewhat incoherent.

3.	The paper lacks analysis on OperA’s training speed and inference latency, which are essential for assessing its practical efficiency.

**Questions:**

	Why does OperA freeze the core tensor? Can OperA’s overall performance truly surpass that of MPOE? This comparison seems to be missing.

	How is the sharing ratio among different components of SharedLoRA determined? It would be helpful to include detailed settings or configuration for this part.

	Is MixLoRA really the state-of-the-art (SoTA) method? Is there any evidence supporting this claim?Several works — such as HydraLoRA, MTL-LoRA, and HMoRA — appear to outperform MixLoRA in their respective scenarios, suggesting that additional baselines should be included.

	Equation (9) seems to contain an error, and the subplots (b) and (c) in Figure 1 are somewhat confusing; revisions are recommended.

---

### Official Review · Reviewer_Qrtu · 2025-10-31

**Soundness:** 3
**Presentation:** 2
**Contribution:** 1
**Rating:** 2
**Confidence:** 4

**Summary:**

Here is the translation using simple vocabulary:

This paper aims to solve problems in PEFT: LoRA adapters are not flexible because their rank is fixed , and MoE uses too many parameters. The paper proposes two new rank-efficient methods: SharedLoRA and OperA. SharedLoRA uses a two-layer structure based on adding parts together, it combines specific adapters with group adapters that are shared. This design uses fewer parameters but achieves a higher effective rank. OperA is a different method inspired by quantum physics that uses multiplication, it uses a technique called MPO , trains only small auxiliary parts, and multiplies them with a fixed center part. This setup uses fewer parameters but achieves an extremely high effective rank

**Strengths:**

- The experiments cover 6 LLMs of different types and sizes (Llama3 series, Phi4, Qwen2.5 series), which matches real-world scenarios .

- The paper lists all hyperparameters in the Appendix , making it easy to reproduce the results.

**Weaknesses:**

- The paper's two methods (SharedLoRA and OperA) seem like two completely separate ideas put together. I don't think these two modules are strongly related.

- There is a lot of past work on shared LoRA and efficient LoRA. This paper does not discuss or compare itself to this past work enough:

https://arxiv.org/html/2410.11772v1

https://arxiv.org/html/2402.08562v1

https://openreview.net/pdf?id=Thv66GmqZS

https://arxiv.org/abs/2406.10785

https://arxiv.org/abs/2410.19694

- In the results (like Table 1), the paper's two methods have their own pros and cons. They are only slightly better than the baseline (MixLoRA). A single, unified solution like MixLoRA seems more useful.

- A basic assumption of this paper might be wrong. The paper claims MixLoRA is the SOTA. However, the original MixLoRA paper only discussed other methods; it never ran experiments to compare against them. There is no proof that MixLoRA is actually the SOTA, which is not a careful claim.


- The paper's methods are limited. They only focus on the LoRA-MoE scenario. This looks fancy, but it is not the main way LoRA is used. Also, combining LoRA + MoE does not give a big performance boost, but it does make inference (running the model) much more expensive.

**Questions:**

Please see weaknesses.

---

### Official Review · Reviewer_jvJ6 · 2025-11-08

**Soundness:** 2
**Presentation:** 2
**Contribution:** 2
**Rating:** 2
**Confidence:** 3

**Summary:**

This paper presents 2 LoRA on MoE finetuning methods. (1) SharedLoRA that has 2-tier of LoRA (2) OperA that is inspired by Matrix product operators (MPO). The goal is to have higher effective ranks without increasing the number of parameters for LoRA. The experiments are conducted on standard commonsense reasoning tasks (e.g. Arc, PIQA, etc) and the main baseline is MixLoRA. The authors on ablation studies show a higher effective rank of OperA than MixLoRA.

**Strengths:**

- The paper is generally easy to follow upon first time of read.

- The experiment setup is correctly designed. The ablation study on effective rank is conducted.

**Weaknesses:**

1. (**critical**) SharedLoRA's novelty is quite incremental. **There is *no* strong motivation behind SharedLoRA and the 2-tier architecture is effectively parameter reuse for non-tier version if I understand correctly.** The effective rank argument for SharedLoRA is (1) naturally holds as we are using less parameters so we expect the higher information usage naturally leads to higher effective rank (2) **on ablation study Figure 2a,b SharedLoRA's middle and last layer behaves similarly as MixLoRA.** The main difference on effective rank is on the first layer which might not matter too much for finetuning (for finetuning middle and last layers are often more important than the first layer).

    - And if I read Table 1 correctly, SharedLoRA only saves 7% parameters compared to MixLoRA with 4 wins and 3 loses on 7 model's finetuning average task accuracy. This result might be on the error margin of initialization, learning rate, etc. **Therefore, I *won't* consider SharedLoRA as having necessarily stronger empirical results than MixLoRA.**

2. (**critical**) I am not familiar with Matrix Product operator (MPO) in quantum physics, but if I read background and Sec 4.2 correctly, MPO is effectively a (high-order) matrix decomposition algorithm.  In this case, we are missing baselines and proper related works discussion for other matrix-decomposition based PEFT in the experiment section.

    - **The discussion on expressivity and the tradeoff between memory-efficiency and inference time are also important** when we consider a more expressive decomposition than a simple low-rank decomposition.

3. Both SharedLoRA and OperA only focus on number of parameters and accuracy while missing proper discussions on other system metrics such as inference throughput. There is always a tradeoff in PEFT community that we can have more expressive PEFT by stacking more operations but at a cost of lower throughput, which make such improvement impractical.

**Questions:**

Although I understand LoRA + MoE is a new direction on PEFT community, we have 1 baseline as MixLoRA in the experiment section which is **too few** to be considered as sufficient comparisons. There are multiple rank-adaptive LoRA variants that we could easily add to MoE in the Table 1 and we should compare them.

---

### Note · Authors · 2025-11-21

I have read and agree with the venue's withdrawal policy on behalf of myself and my co-authors.